# Developing an IoT Framework for Industry 4.0 in Malaysian SMEs: An Analysis of Current Status, Practices, and Challenges

Qusay Adnan Abdulaziz [1,*] , Hazilah Mad Kaidi [1,*] , Maslin Masrom [1] , Halim Shah Hamzah [1],
Shamsul Sarip [1] , Rudzidatul Akmam Dziyauddin [1] and Firdaus Muhammad-Sukki [2,*]

[1] Razak Faculty of Technology and Informatics, Universiti Teknologi Malaysia, Kuala Lumpur 54100, Malaysia
[2] School of Computing, Engineering & The Built Environment, Merchiston Campus, Edinburgh Napier University, 10 Colinton Road, Edinburgh EH10 5DT, UK
* Correspondence: qusay.adnan@graduate.utm.my (Q.A.A.); hazilah.kl@utm.my (H.M.K.); f.muhammadsukki@napier.ac.uk (F.M.-S.)

**Abstract:** This qualitative study aimed to explore the current status, practices, and challenges of Internet of Things (IoT) implementation and to develop an IoT framework for Industry 4.0 in Malaysia. Industry 4.0 enhances a company's manufacturing competitiveness and efficiency. However, the implementation of Industry 4.0 in Malaysian small- and medium-sized enterprises (SMEs) is still in its early stages. Five participants from three different SMEs were selected for online interviews and a focus group. Due to the COVID-19 pandemic, the interviews were conducted online and lasted about 30 to 45 min. The data collected from the interviews were analyzed through thematic analysis and used to validate the literature review and to identify gaps in existing frameworks. The IoT framework was developed through a focus group of experts. This study found that the implementation of Industry 4.0 is relatively low in Malaysian manufacturing SMEs. SMEs are facing various challenges, including the need for education and training, budget constraints, and a lack of experience and knowledge among workers. This study found that the positive impact of IoT implementation included improved internal communication, reduced errors, and enhanced product quality and safety. In addition, this study resulted in the development of an IoT framework for SMEs in Malaysia.

**Keywords:** Industry 4.0; IoT framework; SMEs; Malaysia; manufacturing; status; practices; challenges

## 1. Introduction

Industry 4.0 presents new technological capabilities by integrating emerging ICT technologies to optimize production performance. Through the adoption of Industry 4.0 technologies, the IoT has become increasingly important and visible in manufacturing sectors. However, SMEs often lack the human, technical, and financial resources essential to implement Industry 4.0. Industry 4.0 is defined as a manufacturing procedure that is adapted, service-oriented, optimized, integrated, and interoperable through the use of several advanced technologies [1]. It has revolutionized how things are created, shipped, used, maintained, and serviced [2]. It has also affected the energy footprint of companies as well as their procedures, management, manufacturing power, and skill requirements for supply chain management [3]. The manufacturing industry's steady and continuing flow of brand development programs is driven by shortening product lifecycles [4]. Moreover, the current COVID-19 pandemic presents a chance for a "new generation of entrepreneurs" to drive the next industrial revolution and to develop innovative business models employing cutting-edge technology [5].

According to Bawany [6], Industry 4.0 refers to the concept of "smart factories," in which machines are improved by the use of web connectivity since they are connected to a system capable of seeing the whole manufacturing chain and making choices on its own.

Compared to conventional automated processes, "smart factories" represent an important step toward a fully integrated and flexible system, in which machines and computers interact, collect, and exchange information, and use that data to optimize production efficiency and strengthen a factory's competitive advantage in the market [7].

Because of globalization and speedy technological improvements, small- and medium-sized enterprises (SMEs) now have a better chance of competing successfully. SMEs are recognized as the backbone of the economy, accounting for the majority of a country's gross domestic product [8]. Increased productivity, effectiveness, adaptability, and expanded manufacturing capacity; cost-effectiveness; enhanced quality monitoring and controlling; decreased waste and delivery time; and an incredibly stressful work environment are some of the ways Industry 4.0 might affect manufacturing SMEs [9,10]. By integrating IoT devices and services, SMEs can collect and analyze real-time data from their production procedures, which can help to identify inefficiencies, bottlenecks, and areas for improvement. Then, this information can be used to adjust the behavior of the devices and services in real time, improving overall performance and reducing downtime [11]. Therefore, situation-aware dynamic service coordination in an IoT environment can help to improve efficiency, reduce costs, and can enhance overall productivity [12].

Malaysia's economy relies heavily on SMEs [13]. The majority of SMEs are connected with business organizations, and their contribution to the gross domestic product (GDP) is almost 47%, which, by 2020, expanded to 50% of manufacturing yield [14]. Malaysia's manufacturing industry is the second-largest contributor to overall SMEs, accounting for 5.3% of all SMEs with 47,698 establishments [15]. Since the automotive industry is crucial to Malaysia's efforts to become an industrialized nation, on 21 February 2020, the Ministry of International Trade and Industry (MITI) launched the National Automotive Policy (NAP) 2020 [16], which was an extension and expansion of NAP 2014, that intends to make Malaysia a regional leader in automotive manufacturing. The manufacturing sector of Malaysia contributes to 80% of interregional merchandise trade, while services account for just 20% [17].

The landscapes of neighboring Asian countries such as China and Singapore are also being transitioned by the progress of Industry 4.0. The Singapore Economic Development Board (EBD) has released the Smart Industry Readiness Index to facilitate manufacturers in Singapore to take their first steps towards implementing Industry 4.0 [18]. Similarly, the government of China has introduced the Made in China 2025 initiative, which promotes the industries of a nation into a high-tech mode to protect their position in the international market [19]. Industry 4.0 is just getting started in Malaysia's industrial sectors, which is significant because they are a major source of national income. Malaysian manufacturers may lack information on the precise effects and cost-effectiveness of Industry 4.0-based projects due to the fact that it isa new concept [20].

## 2. Literature Review

### 2.1. Current Status and Practices of Industry 4.0

Through Industry 4.0, the manufacturing sector is changing worldwide from a labor-intensive landscape to a digitalization and automation landscape [21]. Cloud computing, cyber-physical systems, intelligent systems, the Internet of Things (IoT), and robotics are all examples of manufacturing technologies that contribute to the growing trend of automated data sharing and configuration or replanning of production [9]. The goal of Industry 4.0 is to deliver IT-enabled mass customization of manufactured goods; to enable the fully adaptable and automated adjustment of the manufacturing process; to track the components and products; to improve communications between parts, machines, and products; to use human–machine interaction (HMI) paradigms; and to accomplish IoT-enabled manufacturing optimization in smart factories. The aim of industry 4.0 is to establish "smart factories" where the IoT and cloud technology are used to improve and revolutionize industrial technology [22].

Information-sensing devices and systems, the Internet, and various access networks are all part of the IoT, which is a massive intelligent network capable of providing real-time data and insights. Within the Industry 4.0 framework, situation-aware dynamic service coordination can help to improve industrial autonomy and virtualize the production process [12,23]. To be competitive in today's global economy, it is imperative for manufacturers to invest in Industry 4.0 [24]. Furthermore, the manufacturing sector continues to the most significant industry in the economy of Malaysia, as it has the biggest multiplier impact on the country's progress and operations. As of the year 2020, approved investments in Malaysia totaled MYR 164 billion, with the industrial sector receiving the largest share at MYR 91.3 billion [25].

Leaders in international manufacturing industries such as Japan, Germany, and the USA see Industry 4.0 as a chance rather than a risk, although certain manufacturers are significantly less confident due to the complication of altering industry boundaries [26]. Despite that, Shaalan et al. [27] stated that manufacturers are not as ready as they should be for the many features of Industry 4.0 and to reap its benefits. Compared to the United States and Germany, Japan demonstrates a lesser level of readiness for Industry 4.0 [28].

In terms of the industrial revolution, Malaysia's manufacturing sector is currently at a point between Industry 2.0 and 3.0 [29]. The vast majority—98.5%—of Malaysia's 49,101 manufacturing establishments are SMEs, whereas just 1403 are considered to be large firms [30]. The majority of Malaysian industries are SMEs. Currently, SMEs account for 59% of all jobs, and they produced 38% of the country's GDP last year [31]. If SMEs do not realize the critical nature of implementing Industry 4.0 to enhance their manufacturing competitiveness, the country will face a crisis [32].

Only 30% of Malaysian firms are even familiar with Industry 4.0 [33]. Despite widespread recognition among manufacturers of Industry 4.0's potential benefits and chances for increasing competition, the extent of preparation for this shift has varied widely among different nations, industries, and even individual businesses. Executives in Malaysia, however, are largely optimistic as they prepare for Industry 4.0 [34]. According to the Global Competitiveness Index 2017–2018, Malaysia has moved up from the 25th position (2016–2017) to the 23rd position (2017–2018) out of 137 economies around the world in terms of overall competitiveness [35].

Money was set aside in the 2019 federal budget to encourage businesses, especially smaller ones, to implement Industry 4.0 technology and digitize their operations, and therefore compete more effectively in the global marketplace. In the 2019 budget, the Malaysian government allotted MYR3 billion to the "Industry Digitalization Transformation Fund" and MYR210 million to the Readiness Assessment Program to accelerate the uptake of technologies related to Industry 4.0 in Malaysia's industrial sector [36]. As part of the 2020 budget, the Malaysian government planned to invest in the 5G ecosystem, launch the Connectivity Plan, and to provide a range of investment incentives to help local businesses to enter Industry 4.0 [37].

The technological adoption rate in Malaysia is around 20%, notably among SMEs. The majority of manufacturing businesses have adopted less than 50% automation [38], despite measures created by the Malaysian government to support the technological growth of manufacturing firms. While the government of Malaysia has taken steps to encourage SMEs' digitalization, the COVID-19 pandemic has caused only 25% of Malaysian businesses to speed up their digital transformation efforts, while 60% have slowed down. In addition, the Malaysian manufacturing sectors have had difficulties adopting Industry 4.0 due to a scarcity of technical facilities and infrastructure as well as a lack of a highly skilled labor force and resources [39].

Despite the Malaysian government's National Policy of Malaysia on Industry 4.0 (Industry4RWD), which aimed to drive manufacturing industries toward Industry 4.0, the initiatives have not been widely available to the general public at this time because contact is primarily restricted to technocrats. With respect to Industry 4.0, bridging the gap between small- and micro-sized firms or even medium-sized firms at the other end of

the technical divide will be difficult if the necessary support infrastructure is not in place. Production upgrades from 2.0 to 3.0 occur very slowly, which is exacerbated by the fact that Malaysia is not a technology-producing nation [32]. Luthra and Mangla [40] emphasize that Industry 4.0 is a fairy tale to emerging nations when a country is ambiguous on an exact definition of proper practice and understanding in business.

Malaysia ranks above all of the 17 East Asian and Pacific economies, ahead of Indonesia (36), Thailand (32), China (27), and the Republic of Korea (26). Malaysia's manufacturing industry provides 23% of the nation's GDP [41]. Malaysia maintains a competitive and strong position among global competitors despite widespread ignorance of Industry 4.0. Indeed, the Malaysian government has sent alerts to manufacturers about this revolution, and they have been strongly urged to join it. This is a particular challenge for manufacturers who have completely adopted Industry 4.0 standards [42].

### 2.2. Challenges of Industry 4.0 during the Implementation Process

Even though Industry 4.0 would generate economic advantages, there remain challenges to its adoption. The early development of Industry 4.0 is a contentious issue among practitioners and researchers. Numerous scientific, economic, social, political, and technical dimensions may be used to recognize the challenges of Industry 4.0 [43]. During the preliminary phase of implementing Industry 4.0, first, organizations will confront barriers to adopting this new revolution that include acknowledging implementation necessity and preparing the corporation's transition procedures [44].

Before initiating a transition, management has several considerations and concerns regarding the risks associated with the procedures as well as the marginal profits that will result from the adoption of Industry 4.0. These concerns are the impediments to Industry 4.0 adoption. The most likely cause of reluctance among companies preparing to launch an Industry 4.0 effort is the possibility that investment costs may exceed the anticipated growth of a firm, ultimately resulting in an economic deficit [44]. Industry 4.0 is difficult, and its advancement is a building-block procedure. According to a survey conducted by Hamidi et al. [45], one of the challenges faced by firms in implementing Industry 4.0 is a lack of financing for its maintenance.

According to a survey conducted by Müller et al. [46], one of the biggest challenges that SMEs face is the high cost of investing in information technology (IT) infrastructures, machine parts, and, last but not least, the costs for technical training and IT personnel. Improving or replacing old IT infrastructures is expensive, which is a problem for all companies [47]. Massive investments in modern IT infrastructure relevant to Industry 4.0 are required for industrial IoT development [48]. According to Gilchrist's [33] research, companies need financial support to invest in or join Industry 4.0 roadmaps. Further, the preliminary phase of implementing Industry 4.0 has cost implications such as professional consultation, advanced technology through all levels of the organization, high-performance communication and information technologies, infrastructure, highly skilled workforces, and the re-engineering and re-evaluation that will be formed to build the Industry 4.0 platform. Another challenge associated with implementing Industry 4.0 is the uncertainty around its potential financial benefits [49].

Management is hesitant to implement Industry 4.0 due to cost and benefit concerns. Time constraints are another challenge that management faces. The transition to an Industry 4.0 infrastructure is labor-intensive process [50]. Many experts have estimated that it will take 20 years for the manufacturing sector as a whole to make the transition to Industry 4.0 [51]. More specifically, a firm needs a minimum of 10 years to lay out a solid digital foundation [52]. Multiple configurations of smart devices are needed for a full Industry 4.0 platform, which take time and money to build before the platform can be used in manufacturing industries [53]. Regarding managing time, management faces two main challenges. First, the time needed to adopt, test, and execute technologies relevant to Industry 4.0 might have an impact on regular operations. Second, the organization has to

establish a mature Industry 4.0 ecosystem, and the benefits of deploying Industry 4.0 will take some time to become apparent [23].

Management is also concerned with improving the human resource development process to implement Industry 4.0. The availability of skilled personnel at various organizational levels to cope with the rising complexity of future manufacturing procedures is also a significant challenge [54], and it is not exclusive to the financial capital necessary to install complicated technology. In the future, autonomous robots will take over low-skilled employment, forcing workers to acquire new competencies connected to the use of smart technologies [55]. According to research on the risks of Industry 4.0 conducted by Tupa et al. [56], human resource (HR) departments are likely to have a shortage of suitably trained employees for deployment in the new digital workplace. The manufacturing problem will worsen due to a lack of specialists and people willing to take on steep learning curves. The lack of experience and workers on learning slopes will worsen the manufacturing issue [57].

Workers' jobs will soon be increasingly automated; thus, it is important to equip the next generation with the skills necessary for Industry 4.0 [58]. According to a survey by Müller et al. [46], getting staff on board with Industry 4.0 is a major challenge for businesses. The lack of Industry 4.0 capabilities and skills, mainly in the area of IT, is the reason for the low acceptance of Industry 4.0 competencies and skills, leading to job loss. However, rather than being replaced by robots, Bragan et al. [59] see humans and machines working together. An important aspect of a worker's role in the age of Industry 4.0 is to manage complicated and indirect activities, such as monitoring everyday activities using smart devices instead of transferring goods. Therefore, management should place a premium on multidisciplinary training in areas such as engineering, informatics, mathematics, and economics [48].

There is a potential for several psychological risks to emerge, which poses a serious challenge to occupational safety and health [57]. For instance, existing workers may experience unease when manual labor is replaced by automation or robots [59]. Indirect risks rise when humans and technology work closely together. In the context of Industry 4.0, Khalid et al. [60] listed the potential risks associated with human and machine collaboration, including those posed by the robot itself, those posed by the industrial process, and those posed by a malfunctioning robot control system. The success of cyber intelligence relies heavily on computational intelligence for tracking, analyzing, and recognizing virtual potential threats to combat hackers, viruses, and terrorists who use the World Wide Web (www) for cyber harassment and stalking coercion, extortion, stock market deception, complicated industrial espionage, and making plans or carrying out terrorist activities. With respect to optimizing production and manufacturing decisions, today's information-based network construction lacks the necessary self-computing, self-sensing, self-maintaining, and self-organizing capabilities.

Ling et al. [23] described the numerous challenges that have been identified as continuing to be technical challenges in the early stages of Industry 4.0, especially the growth of the IoT, such as the absence of standardization, the reality that privacy and security solutions are needed, and the errors in data analytics technologies. Most firms are concerned about the privacy and security of their data since the infrastructure of Industry 4.0 is either cloud-based or web-based [61]. Cybersecurity is the problem, as information has become a lethal resource [62]. Massive amounts of data are collected from many sources, for instance, by evaluating manufacturing data and integrating the findings with consumer information systems [47]. In addition, firms in the corporate value chain should be able to send and receive data via the IoT in an ideal Industry 4.0 setting. Confidential data and information may be leaked because various cooperating firms have varied levels of cybersecurity. Small firms within a value chain with limited resources may have inadequate cybersecurity systems [63].

Despite a firm's full adoption and integration of all Industry 4.0-related technologies, communication protocols, and IoT hardware [64], it is claimed that there is still a gap

for this completely linked platform to gather relevant data and to analyze that data for implications. There is concern over cybersecurity due to insufficient technical support for the development of Industry 4.0 technologies as well as the leakage of sensitive data through the internet. To stop cyberattacks on the IoT data connection and data theft, all participants in the value chain must take precautions to protect their data [65].

The use of the IoT, people, and data will generate opportunities for industrial espionage, data theft, and cyberattacks [47]. The cybersecurity problems are not just restricted to data disclosure but are on the edge of betraying private data, and last but not least, the production system is under the unlawful control of third parties [26]. Industry 4.0 will increase the need for secure data architecture and secure design practices [66]. Automatic detection of viruses, attacks, and threats with zero installation is a must-have for the smart manufacturing system. Meanwhile, Gilchrist [33] investigated the sustainability challenge that many businesses face when attempting to implement Industry 4.0 across their entire organization and all of their factories at once. Up until now, many businesses relied on a separate application network. When several real-world items are linked together, scalability becomes a challenge. The challenge of scalability is complicated by the need for multi-tiered data transfer and communication, data management and processing, and service delivery [67]. To meet the need for whole-company and cross-company collaboration, as well as that of suppliers and consumers, a scalable solution is necessary [68].

A report by Deloitte [47] pointed to one of the main challenges of Industry 4.0 as being the management of a vast amount of generated data and the transformation of the data into meaningful information. According to a survey conducted by Thames and Schaefer [69], the integration of data is one of the top three challenges encountered during the implementation of Industry 4.0. The difficulty in integrating data from various smart devices into a standard format was also highlighted as a challenge to Industry 4.0 [68]. In Industry 4.0, data become a company's most valuable assets; therefore, the risks associated with data collection, interpretation, and disclosure are very high. Information risks, including data loss, integrity loss, and a lack of relevant information, can occur during management system implementation [70]. New product or service launches, innovations, and alternative business models can all pose challenges if a firm is having problems with technology. Different types of machines have variable degrees of autonomy and have different lifecycles that put a strain on businesses because some equipment needs to be replaced while others may need modification to different degrees [26]. The extent to which a business adopts Industry 4.0 technologies is influenced by the size of the firm [70].

SMEs have an advantage when it comes to speeding up the digital revolution since they can more readily design and implement new IT structures. In contrast to organically evolved structures, however, multinational companies and large companies have more complexity to worry about. Large firms that practice regular process management will see the benefits of Industry 4.0 implementation sooner than SMEs [71]. Unless new systems replace the present ones with manageable effort and justified risk, these two types of firms must guarantee that new technologies are compatible with their current IT infrastructure systems [47]. However, compared to large firms, SMEs have fewer resources to deal with technical, financial, and human resource challenges [72].

The most significant challenges that SMEs face when embracing Industry 4.0 are the ability to develop a workable Industry 4.0 approach, the shortage of financial resources, a low degree of standardization, the cost and benefit analysis of Industry 4.0 technologies, unawareness of the integration concept, data security, and a lack of skill in IT [70,73]. According to Zaidi and Belal [74], Malaysian SMEs choose to be technology followers since they can only learn more about IoT-related products and services with the help of a third party. As a result, the technical challenge is not dominating in Industry4RWD. However, it is difficult for technology, software, and system developers to create a new and effective Industry 4.0 platform. Immature technology may interpret incorrect data extraction or interpretation.

The lack of understanding Industry 4.0 implementation, particularly the IoT, hinders SMEs from effectively contributing to the country's economy [23]. The barriers to the successful implementation of Industry 4.0, and particularly the IoT, have not been sufficiently researched, especially in the Malaysian context [75]. Furthermore, there is a lack of a proper framework for Industry 4.0 implementation, particularly for the IoT, which makes it difficult for Malaysian SMEs to successfully implement Industry 4.0 technologies in their production lines [23]. Therefore, this study aims to fill these gaps by identifying the barriers hindering SMEs from successful Industry 4.0 implementation and by developing an IoT framework to support Industry 4.0, using Company (A) as the main case study.

The contributions of this paper are as follows:

1. What are the current practices of the IoT/Industry 4.0 in Malaysian SMEs?
2. What are the barriers to IoT/Industry 4.0 in Malaysian SMEs?
3. What components in a specific IoT framework can be used to support Industry 4.0 implementation in a Malaysian SME stamping production line?

### 3. Materials and Methods

Figure 1 shows the flow chart of this research. It was carried out using a qualitative method through online interviews and a focus group. The qualitative method provides insights into research objectives through primary data and helps to improve the ideas or hypotheses of a new framework's implementation [76]. Therefore, this study adopted the qualitative method. Five participants, representing various positions within three different SMEs, were chosen as respondents. These individuals included a plant manager, two project managers, the CEO, and an information technology officer. These participants were selected based on their extensive experience in the industry and their ability to provide insightful information about IoT implementation in SMEs. The first phase was focused on achieving the first two objectives of this study. The sample for the research study is presented in Table 1, including relevant information about the participants such as department, company, current position, and years of experience. In the first phase of the research, two parts were conducted using the interview method. For each part, five interviews were conducted with the same participants. The participants were selected based on their expertise and experience in the fields of the IoT and Industry 4.0 in Malaysian SMEs. The second phase of the research was conducted using the focus group interview method. Only one interview was conducted with participants from one company (A). The focus group interview allowed for the exploration of collective opinions and experiences, thus providing a deeper understanding of the phenomenon [77].

**Table 1.** Study sample.

| Participant | Department | Company | Current Position | Years of Experience |
|---|---|---|---|---|
| Participant 1 | Operation | A | Plant/Operation Manager | 28 |
| Participant 2 | Maintenance | A | Head of Maintenance | 20 |
| Participant 3 | Operation | A | Innovation Manager | 19 |
| Participant 4 | IT/Development | A | Project Manager/Software Developer | 14 |
| Participant 5 | Automation | A | Robotics Manager | 11 |

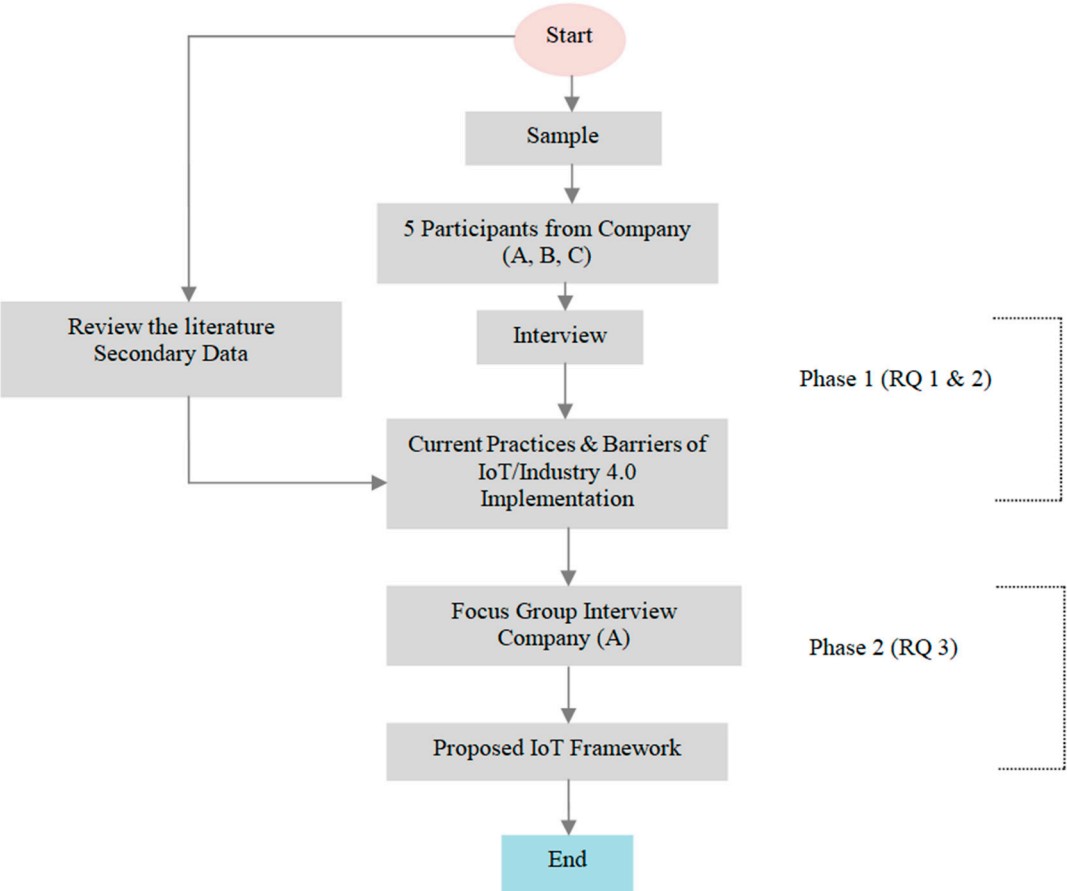

**Figure 1.** Flowchart.

Due to the ongoing COVID-19 pandemic, face-to-face interviews were not possible. Instead, the researcher arranged for online meetings through the Zoom platform, which lasted approximately 30 to 45 min for each interview. According to O'Connor and Madge [78], online interviews offer greater convenience and cost-effectiveness compared to face-to-face interviews, with participants enjoying the flexibility to choose the time and location of their participation. To ensure the validity of the interviews, a pilot test was conducted with a sample size of two participants to assess the relevance, clarity, and appropriateness of the questions. Pilot testing is a small-scale study carried out before starting an actual experiment, which is intended to test and refine methods [79]. Based on the results of the pilot test, the questions were modified to suit the positions and expertise of participants, thus increasing the validity of the collected data. To ensure the reliability of the interview data, a structured approach was used, and interviewers were trained to ask questions consistently and to avoid leading suggestions. Further, the responses were recorded and analyzed to uncover key themes and insights. The data collected from these interviews were used to validate the findings of the literature review and to identify gaps or limitations in existing frameworks.

The data collected from the interviews were analyzed through the NVivo Version 12 software using thematic analysis to identify themes and patterns in the responses. During the second phase of the research, the focus was on developing a framework for the IoT. To ensure the accuracy and quality of the data collected, a focus group was employed. Six participants, consisting of experts in the fields of the IoT and Industry 4.0, were selected to participate in the focus group session, which was held through a Zoom meeting. The responses collected from the participants were analyzed and used to develop the IoT framework. The name of the company was hidden to maintain confidentiality and to respect ethical considerations regarding privacy. Figure 1 presents a clear and concise

visual representation of the research materials and methods utilized in the study, providing a comprehensive overview of the study design and procedures.

## 4. Results

### 4.1. Perceptions about IoT and Industry 4.0

When the participants were asked, "What is IoT?", the majority of the participants perceived the IoT as "a method of communication and interaction between machines and humans where devices can be programmed to perform specific actions remotely using IoT technology" and "exchanging information across a network of interconnected machines using the internet". All participants perceived that the IoT involves the connection and exchange of data between various machines through the internet. Participant 1 focused on the "programming" aspect of the IoT, while Participant 2 focused on the "exchange of information". Participant 3 emphasized the "value companies can gain through interconnected devices" and Participant 4 defined IoT as "anything connected to the internet". Participant 5 highlighted the "remote management and monitoring aspects of the IoT". When the participants were asked, "What is Industry 4.0?", the majority of participants perceived Industry 4.0 as involving automation and the integration of technology, such as big data, IoT devices, and real-time communication. The perception that stood out as different was that of Participant 2, who defined Industry 4.0 as a "digital link between a server and local networks that provides real-time data".

### 4.2. Current Status and Practices of the IoT and Industry 4.0

When the participants were asked, "Has your company implemented IoT/Industry 4.0?", the majority of the participants reported that their company is either in the discussion stage or has taken limited steps toward the implementation of the IoT and Industry 4.0. Participants 2 and 3 reported that "their departments are in the discussion stage or have not yet taken serious steps towards implementation, and that they do not plan to implement IoT/Industry 4.0". Participant 3 reported that "they are using computer-based technology but only for manual data entry and do not plan to implement IoT/Industry 4.0 as they feel it's too late to transform". Participant 5 reported that "their company has implemented the SAP system but has not yet fully adopted IoT/Industry 4.0".

When the participants were asked, "How did the top management prepare the employees to implement IoT/Industry 4.0?", the majority of the participants believe that the top management has taken steps to prepare employees for IoT and Industry 4.0 implementation. Participants 1, 3, and 5 reported that "the management has hired technology experts, sent employees for training, regularly updated employees, and is actively engaged in preparing employees through various methods and meetings". However, Participant 2 reported that "they have not yet made any plans for employee preparation". In addition, Participant 4 emphasized the importance of "practical training on actual machines to improve employees' technical knowledge".

When the participants were asked, "Is the top management of your company eager to learn about IoT/Industry 4.0?", the majority of the participants reported that their top management is actively interested in learning about the IoT and Industry 4.0 and holds regular meetings to discuss it. However, Participant 2 reported that "their company is not currently planning to learn about IoT/Industry 4.0, and discussions related to new technology are limited". Participant 3 reported that "their top management discusses the transformation they will bring to the company".

When the participants were asked, "How much money has your company spent on technology upgrades between 2017 to 2021?", the majority of the participants reported spending between MYR250,000 and MYR1,400,000 on technology upgrades between 2017 and 2021. However, Participant 2 revealed that "their company spent MYR 1,400,000 on purchasing new devices, but they remain unused for the intended purpose". While Participant 4 reported that "their company invested close to MYR1,000,000 on new machinery and started to see benefits from the investment immediately" and Participant 5 shared that

"they spent MYR250,000 on a new high-frequency welding machine but not on Industry 4.0 or IoT technology". The participants have different experiences with their technology investments, with some seeing immediate benefits and others having unused devices.

### 4.3. Barriers to the Implementation of the IoT and Industry 4.0

When the participants were asked, "What are the barriers your company is likely to face when implementing IoT/Industry 4.0?", the participants generally agree that the barriers to implementing Industry 4.0 and the IoT in their company include education and training for staff, budgeting, and financial management. Participants 1 and 2 both mentioned "the need for employee training as a barrier". Participant 3 highlighted "securing sufficient funding as a barrier" while Participant 4 added "a lack of experience and knowledge among workers". Participant 5 emphasized "securing a budget for implementation and maintenance as the main barrier".

When the participants were asked, "What does your company policy say about IoT/Industry 4.0? Is it likely to affect the implementation of Industry 4.0?", the majority of participants view the company's policy on the IoT and Industry 4.0 positively or with room for improvement. However, Participant 2 was dissatisfied with the company's existing policy and stated that "it does not provide clear guidelines on the implementation of new technology and it will negatively impact the company". Participant 5 expressed concern about the lack of any state of IoT/Industry 4.0 in their company's policy and believed that "implementing new technology without preparation will harm the company's goal of becoming the best automotive vendor in the country".

When the participants were asked, "When faced with IoT/Industry 4.0 problems, what do you do about them? Is management supportive enough to help solve your problems?", the majority of participants reported that management helps to resolve Industry 4.0 and IoT issues and is working to improve the technology. Only Participant 2 reported "administrative issues due to aging servers and inadequate networking infrastructure" while Participant 5 reported that "the company has yet to implement IoT/Industry 4.0". Participant 4 reported that the company uses a "predictive path method" to collect and analyze machine data to predict faults and resolve problems promptly.

When the participants were asked, "What is the biggest challenge you have faced about the implementation of IoT/Industry 4.0? How did you and your company overcome it?", the majority of participants had faced challenges related to funding and the availability of skilled staff. To overcome these challenges, they sought financial assistance from the government, initiated implementation on a small scale, and provided training to employees. Participant 3 faced a unique challenge related to "unstable device connections due to poor IT infrastructure, which their company is addressing by installing a high-performance IT network". Participant 5 faced additional challenges related to "material supply and supply chain during COVID-19, as well as hurdles related to funding, a skilled workforce, and precise planning, which many SMEs must focus on addressing".

### 4.4. IoT Implementation Framework

When the participants were asked, "What technologies are available presently at the company?", the focus group participants reported that the "company is currently working towards implementing IoT by incorporating robotic technology". The aim is to leverage advanced technology to improve operations. However, the company is facing challenges in effectively integrating the robots and devices using the IoT due to a lack of skills and knowledge. The company is still learning how to improve communication between devices through the IoT. One participant stated that "the company has the necessary robots and advanced machines but requires adequate training and guidance to utilize them to their full potential". The focus group participants also noted that "the company has installed centralized servers at various locations within the plant, allowing for easier connectivity to the network for new devices". The company has also established a central operations room to monitor data from multiple sources. Currently, the company can view "real-time data,"

but plans to expand its capabilities by processing the data in the future. "The company has the goal of achieving Industry 4.0 by incorporating more cutting-edge technologies into its facilities".

When the participants were asked, "Do you ever follow any best practices or protocols to streamline your production?", the focus group participants stated that "the company implements various practices and protocols to streamline production processes. They follow a communication and production protocol and have adopted the Lean production system from Toyota to improve communication among departments". However, they acknowledge that "there are challenges associated with the Lean system and plan to integrate IoT protocols to address these issues". The company also follows "a standard process flow, which is presented during audits and covers the material flow from the warehouse to final assembly". However, the group noted that "this protocol system, influenced by the Perodua production system, lacks accuracy and can be time-consuming due to manual reporting".

When the participants were asked, "Have you implemented any IoT systems in your company? If yes, what is your expertise on the implementation side of the IoT?", the focus group participants highlighted that the company had successfully implemented IoT in two of their production line machines, and as a result, they were able to monitor outputs in real time and improve efficiency. They noted that "the company has reduced the traditional way of manually performing tasks and enforced monitoring of the production system to increase efficiency". The group also pointed out the benefits of the IoT, stating that it "should be utilized for easier and more efficient planning of the delivery of products or receiving orders from customers" and that they "are still not benefiting from many other features of IoT and the company should focus on adopting more of these technologies".

When the participants were asked, "How were these technologies implemented? Have you come across any framework that can be used to help make IoT/technology implementation smoother?", the participants of the focus group mentioned that "they have adopted the ISO/TS standard designed especially for the automotive industry". The group highlighted the significance of the standard as "it provides an industry framework to achieve best practices in designing and manufacturing products for the automotive supply chain". They referred to the standard as the "workflow" and noted that it serves as "an important communication flow". The group mentioned that "if it is used in the implementation of Industry 4.0, it will make the process simpler and provide clear evidence of industry 4.0 successes in the company". Additionally, the standard gives a "systematic information flow based on operational data to get better results".

When the participants were asked, "Can the same blueprint or framework be used to implement Industry 4.0 or IoT technologies at the company?", the focus group response indicated that "the current framework or standard can be adapted to implement Industry 4.0 or IoT technologies" as long as "the company enhances its capabilities or performs necessary modifications to make it compatible with these technologies". The participants noted that with some modifications to align with IoT and Industry 4.0 requirements, the company will have a clear path for successful implementation. Further, they highlighted that "the existing framework or standard has already shown positive results and improved performance, and the company has good experience running it, making it more cost-effective and easier to utilize it for upcoming technologies instead of adopting a new framework or standard".

When the group participants were asked, "If you are asked to develop a framework that can be used to help make IoT/technology implementation smoother, what would the framework look like?", a framework for smoother IoT and technology implementation was proposed by the group participants who stated that the "company will order from the suppliers first, and then the materials will go to the store from where it will proceed for the production". After that, "material will go to WIP (work in progress), where the final product will be sent to the warehouse". The important thing is "controlling and communicating information regarding each stage effectively to the other". The group emphasized the

benefits of implementing IoT in the framework, stating that it will "provide better accuracy in calculations, enhance communication, and make the process faster and simpler". On the one hand, according to the group, "the manual practices involve many people to authorize the progress on each step, which wastes a lot of time, produces multiple errors, and is very disadvantageous in case of an emergency". On the other hand, the implementation of IoT technology in the framework "eradicates limitations of time and place, enabling the industry to be monitored and controlled remotely". The group mentioned that "in the proposed framework integrated with IoT technology, the information of the activities at each stage of the material flow would be shared in real time and the process would work according to it".

When the participants were asked, "What is the expected general impact on the performance of SMEs when implementing the technological framework, especially in IoT?", the focus group participants mentioned that "the implementation of the technological framework/IoT is expected to have a positive impact on SMEs' performances". The group highlighted that "it will improve internal communication, reduce errors, and enhance product quality and safety". The success of this framework will also "impact the company's business," making it a "reliable player in the market with a high-quality reputation, leading to more customer orders". Customization opportunities through the framework will "give the company an edge over others". However, the group recommended that "Malaysian SMEs should not limit themselves to the national level but expand globally to compete with other SMEs". With effective marketing, "they can showcase their capabilities and the quality of their products on an international level". The following framework (see Figure 2) was proposed on the base of the above study results.

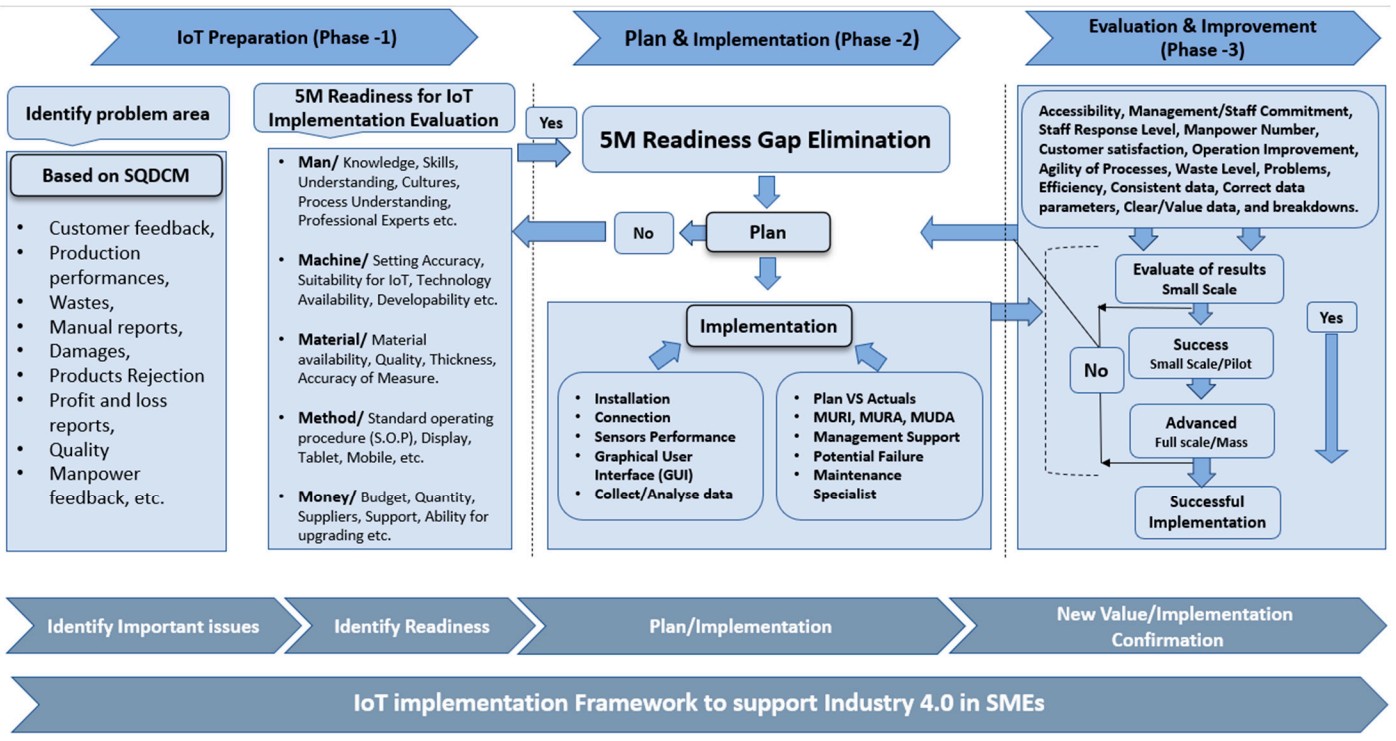

**Figure 2.** IoT implementation framework to support Industry 4.0 in SMEs.

The above framework can be used to support the implementation of IoT in real-world scenarios, particularly in SMEs. The three phases, i.e., IoT preparation, planning and implementation, and evaluation and improvement provide a structured approach to ensuring that IoT implementation is successful and sustainable. The three phases are described below.

### 4.4.1. IoT Preparation (Phase1)

During this phase, the organization needs to evaluate its readiness for IoT implementation using the 5M model (man, machine, material, method, and money) based on SQDCM (safety, quality, delivery, cost, and morale). The organization needs to assess its accessibility, management/staff commitment, staff response level, manpower number, customer satisfaction, operation improvement, the agility of processes, waste level, problems, efficiency, consistent data, correct data parameters, clear/value data, and breakdowns. This assessment helps to identify the problem areas and the organization's readiness for IoT implementation.

### 4.4.2. Planning and Implementation (Phase2)

During this phase, the organization needs to plan and implement the IoT system. Planning should be based on the evaluation of results on a small scale and should identify important issues and eliminate the readiness gap. The organization needs to assess the 5M readiness for IoT implementation evaluation. The implementation should focus on machine/setting accuracy, suitability for IoT, technology availability, developability, material availability, quality, thickness, the accuracy of measurements, standard operating procedures (SOPs), display, tablet, mobile, etc. Implementation should also consider the graphical user interface (GUI) for data collection and analysis.

### 4.4.3. Evaluation and Improvement (Phase3)

During this phase, the organization needs to evaluate the implementation results and to identify areas for improvement. The organization needs to assess customer feedback, production performances, wastes, manual reports, damage, product rejection, profit and loss reports, quality, manpower feedback, etc. The evaluation should focus on planning vs. actual, muri (overburden), mura (unevenness), muda (waste), and management support. The organization needs to assess potential failure, maintenance specialists, installation and connection, and sensors performance. Successful implementation confirms the new value/implementation.

## 5. Discussion

Based on the study results, it was found that SMEs have an understanding of the IoT and Industry 4.0. The majority of participants perceived it as a method of communication and interaction between machines and humans, where devices can be programmed to perform specific actions remotely using IoT technology and the exchange of information across a network of interconnected machines through the internet. Different participants highlighted different aspects of the IoT, such as programming, the exchange of information, the value gained from interconnected devices, anything connected to the internet, and remote monitoring and management. Similarly, the majority of participants perceived Industry 4.0 as involving automation and the integration of technology such as the IoT devices, big data, and real-time communication. These findings demonstrate that SMEs have an understanding of the IoT and Industry 4.0, which is significant for the successful implementation of these technologies in their businesses. The results of this study contradict the findings of previous studies [70,73], which have shown that the primary obstacle faced by SMEs during the implementation of Industry 4.0 and the IoT is a lack of knowledge. Only 30% of Malaysian firms are familiar with Industry 4.0 [33].

The study results showed that SMEs have either taken limited steps or are in the discussion stage of implementing the IoT and Industry 4.0. In regard to top management's preparation of employees for the implementation, the results showed that management has taken steps to prepare employees by hiring technology experts, sending employees for training, regularly updating employees, and actively engaging in preparation through various methods and meetings. The results are in line with those of a previous study [38], which showed that the adoption rate of technology, particularly among SMEs in Malaysia, was around 20%, with most manufacturing businesses implementing less than 50% au-

tomation. In terms of top management's interest in learning about the IoT and Industry 4.0, the results indicated that the top management of SMEs is actively interested and holds regular meetings to discuss it. In terms of technology upgrades between 2017 and 2021, this study highlights that SMEs were spending between MYR 250,000 and MYR1, 400,000. Despite the cost, financing for maintenance remains to be a challenge, as previously noted by Hamidi et al. [45]. However, experiences with technology investments vary, with some SMEs seeing immediate benefits and others having unused devices. There is still an element of uncertainty surrounding the potential financial advantages of the IoT in Industry 4.0, as reported by Koch et al. [49]. The results indicate a range of experiences with IoT and Industry 4.0 implementation and technology upgrades among the participants, which reflects the diverse nature of SMEs in Malaysia.

The results of this study indicated that the challenges associated with the IoT and Industry 4.0 implementation in SMEs are related to education and training for staff, budgeting and financial management, and a lack of experience and knowledge among workers. These challenges can be classified under the scientific, political, social, technical, and economic dimensions of Industry 4.0 [43]. According to Müller et al. [46], the high cost of IT infrastructure and the required materials are major challenges for SMEs, as a significant investment is necessary for the development of the industrial IoT [48]. Most participants view their company's policy on the IoT and Industry 4.0 positively, but some participants have concerns about the lack of clear guidelines or the absence of a state of the IoT and Industry 4.0 in their company's policy. Regarding resolving Industry 4.0 and IoT problems, it was found that management was supportive and was working to improve the technology. However, some SMEs face challenges such as administrative issues due to aging servers and inadequate networking infrastructure or material supply and supply chain issues during the COVID-19 pandemic. The greatest barriers faced by the majority of SMEs are related to funding and the availability of skilled staff, which they have addressed by seeking financial assistance from the government, initiating implementation on a small scale, and providing training to employees. The findings of Tupa et al. [56] emphasized a similar issue, where a shortage of skilled workers for IT adoption was identified as a major challenge for SMEs. Some SMEs have also faced challenges related to unstable device connections and precise planning, which are important for SMEs to address.

The results regarding the implementation of the IoT and Industry 4.0 in SMEs reveal a mixed picture. On the one hand, the SMEs have already been working towards IoT implementation by incorporating robotic technology to leverage advanced technology to improve operations. On the other hand, the SMEs are facing challenges in effectively integrating robots and devices using the IoT due to a lack of skills and knowledge. The SMEs have installed centralized servers and established central operation rooms to monitor data from multiple sources, but they still require adequate training and guidance to fully utilize the robots and advanced machines. The SMEs are striving towards Industry 4.0 by incorporating more cutting-edge technologies into their facilities. The SMEs have implemented various practices and protocols to streamline production processes such as following a communication and production protocol, adopting the Lean production system from Toyota, and following a standard process flow influenced by the Perodua production system. The SMEs acknowledge the challenges associated with the Lean system and the lack of accuracy in the standard process flow and plan to integrate IoT protocols to address these issues. The SMEs have successfully implemented the IoT in two of their production line machines, which has resulted in increased efficiency by reducing manual tasks and enabling real-time monitoring. However, the SMEs are not benefiting from many other features of the IoT and should focus on adopting more of these technologies.

The SMEs have adopted the ISO/TS standard designed specifically for the automotive industry as a framework for IoT technology implementation. This study highlighted the significance of the standard as it provides an industry framework to achieve best practices in designing and manufacturing products for the automotive supply chain and serves as an important communication flow. This study proposes that the current framework or

standard can be adapted to implement Industry 4.0 or IoT technologies as long as the SMEs enhance their capabilities or perform necessary modifications to make them compatible with these technologies.

This study proposed a framework for smoother IoT and technology implementation that involves ordering materials from suppliers, storing them, and then proceeding with production. The IoT framework developed for Industry 4.0 in Malaysian SMEs is comprised of three phases: IoT preparation (Phase 1), planning and implementation (Phase 2), and evaluation and improvement (Phase 3). During the first phase, the focus is on preparation, identifying the 5M readiness for IoT implementation, and eliminating any gaps in manpower, knowledge, skills, understanding, and processes. During the second phase, the focus is on the actual implementation, which involves machine and material preparation, method and graphical user interface (GUI) development, data collection and analysis, and planning and implementation. The final phase focuses on evaluating the results and making improvements based on the evaluation of customer feedback, production performances, waste levels, manual reports, profit and loss reports, quality, manpower feedback, and more. The framework is designed to support Industry 4.0 in SMEs and is evaluated by comparing the actual results with the plan. The potential for failure and the need for maintenance specialists are also considered, and a successful implementation is confirmed through new value and implementation confirmation. The framework emphasizes the importance of controlling and communicating information effectively at each stage of the material flow. The implementation of IoT technology in the framework would eradicate limitations of time and place, enabling the industry to be monitored and controlled remotely, providing better accuracy in calculations, and enhancing communication. The proposed framework can be used practically by organizations to assess their readiness for the IoT, to plan and implement an IoT system, to evaluate the implementation results, and to continuously improve the system. By leveraging this framework, organizations can leverage IoT technology to improve their operations, profitability, and customer satisfaction.

## 6. Conclusions

In conclusion, SMEs in Malaysia have a general understanding of Industry 4.0 and IoT and are taking steps toward their implementation. However, SMEs are facing various challenges such as the need for education and training, budget constraints, and a lack of experience and knowledge among workers. To overcome these challenges, SMEs have sought financial assistance from the government, provided training to employees, and initiated implementation on a small scale. The studied SMEs have successfully implemented the IoT in some production lines and have adopted the ISO/TS standard as a framework for IoT implementation. Therefore, this study proposes a framework for IoT implementation that emphasizes proper management and communication of information throughout the material flow, which can help SMEs overcome challenges in implementing IoT and Industry 4.0 technologies. Implementation of the proposed IoT framework eliminates the constraints of time and location, allowing for remote monitoring and control of the industry.

Therefore, it is suggested that SMEs should focus on incorporating more IoT technologies to fully benefit from their features and should consider the proposed framework modification to ensure they are compatible with Industry 4.0 and IoT technologies. Further, SMEs can address the challenges highlighted in this study and retain skilled staff by providing training opportunities, financial assistance from the government, and initiating implementation on a small scale. SMEs should carefully plan their budget and financial management to effectively implement IoT and Industry 4.0 technologies. This study framework has some implications as it can help Malaysian SMEs to improve their manufacturing competitiveness and efficiency, overcome challenges in implementing IoT and Industry 4.0 technologies, and drive innovation and growth in the manufacturing sector. The theoretical implications of this research are that it contributes to the existing literature on Industry 4.0 and IoT implementation. Furthermore, the proposed framework can help to establish new theoretical models and frameworks for future research on the

implementation of Industry 4.0 technologies in the industrial/manufacturing sector. This study has some limitations. One limitation of the study is the small sample size comprised of five participants from three different SMEs, which may not be representative of the entire SME population in Malaysia. Additionally, the interviews were conducted online due to the COVID-19 pandemic, which may have affected the quality of the data collected compared to in-person interviews. Finally, the study's reliance on thematic analysis may have introduced researcher bias in the data analysis process. Future research is needed to validate the proposed framework. Further, future studies should explore the skill requirements, access to financing, the role of government support, and its impact on promoting IoT and Industry 4.0 adoption among SMEs.

**Author Contributions:** Conceptualization, Q.A.A., H.M.K. and M.M.; data curation, Q.A.A.; methodology, Q.A.A. and H.M.K.; software, Q.A.A.; validation, H.M.K., M.M. and S.S.; formal analysis, Q.A.A. and H.M.K.; investigation, Q.A.A.; resources, H.M.K.; writing—original draft preparation, Q.A.A. and H.M.K.; writing—review and editing, Q.A.A., H.M.K., M.M., H.S.H., S.S., R.A.D. and F.M.-S.; visualization, Q.A.A. and F.M.-S.; supervision, H.M.K., M.M. and H.S.H.; project administration, H.M.K. and F.M.-S.; funding acquisition, H.M.K., M.M., H.S.H., S.S., R.A.D. and F.M.-S. All authors have read and agreed to the published version of the manuscript.

**Funding:** This research did not receive any external funding.

**Institutional Review Board Statement:** Not applicable.

**Informed Consent Statement:** Informed consent was obtained from all subjects who participated in the interview.

**Data Availability Statement:** The data used in this study are available upon request from the first corresponding author, Q.A.A.

**Conflicts of Interest:** The authors declare that they have no conflicts of interest that would influence the validity of the present research.

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
