# Peer review of "Developing an IoT Framework for Industry 4.0 in Malaysian SMEs: An Analysis of Current Status, Practices, and Challenges"

_applsci, doi:10.3390/app13063658_

Round 1

Reviewer 1 Report

The paper represents a developing an IoT framework for industry 4.0 in Malaysian, and found that implementation of Industry 4.0 is relatively low in Malaysian manufacturing SMEs. 21 SMEs are facing various challenges, including the need for education and training, budget 22 constraints, and a lack of experience and knowledge among workers. Also found the positive 23 impact of IoT implementation including improved internal communication, reduced errors, and 24 enhanced product quality and safety. The paper fits into the scope of the journal and its topic in general is interesting for the readers.

1. The paper focuses on a timely and relevant problem and some of the descriptions and proposed methods can be potentially interesting; however, the work lacks focus and several related work in IoT service scenarios, such as ”situation-aware dynamic service coordination in IoT environment”.

2. The proposed developing an IoT framework for industry 4.0, but it is not described in the paper how the system can work in a real environment.

3. In the performance section. The more details of the performance result should be analyzed and provided.

Author Response

Dear Reviewer 1.

We appreciate your time and effort to review our paper. The following are the response to your queries.

  • The paper focuses on a timely and relevant problem and some of the descriptions and proposed methods can be potentially interesting; however, the work lacks focus and several related work in IoT service scenarios, such as ”situation-aware dynamic service coordination in IoT environment”.

Response: Thank you for your comment. We made some changes to the paper to address this issue (see Page 2 second Para. and Page 3 first Para).

".....By integrating IoT devices and services, SMEs can collect and analyze real-time data from their production procedures, which can help identify inefficiencies, bottlenecks, and areas for improvement. This information can then be used to adjust the behavior of the devices and services in real time, improving overall performance and reducing downtime [11]. So, situation-aware dynamic services coordination in an IoT environment can help improve efficiency, reduce costs, and enhance overall productivity [12]."

"Information-sensing devices and systems, the Internet, and various access networks are all part of the IoT, which is a massive intelligent network capable of providing real-time data and insights. In the Industry 4.0 framework, situation-aware dynamic services coordination can help improve industrial autonomy and virtualize the production process [12, 23]. ...."

  • The proposed developing an IoT framework for industry 4.0, but it is not described in the paper how the system can work in a real environment.

Response: Thank you for your comment. We made some changes to the paper to address this issue (see Page 12 last Para. and Page 13 first half page, also in Discussion and Conclusion).

"The above framework can be used to support the implementation of IoT in real-world scenarios, particularly in SMEs. The three phases i.e. IoT preparation, plan and implementation, and evaluation and improvement provide a structured approach to ensuring that the IoT implementation is successful and sustainable. The three phases are as follows:

4.4.1. IoT Preparation (Phase -1):

In this phase, the organization needs to evaluate its readiness for IoT implementation using the 5M model (Man, Machine, Material, Method, and Money) based on SQDCM (Safety, Quality, Delivery, Cost, and Morale). The organization needs to assess its accessibility, management/staff commitment, staff response level, manpower number, customer satisfaction, operation improvement, the agility of processes, waste level, problems, efficiency, consistent data, correct data parameters, clear/value data, and breakdowns. This assessment helps in identifying the problem areas and the organization's readiness for IoT implementation.

4.4.2. Plan & Implementation (Phase-2):

In this phase, the organization needs to plan and implement the IoT system. The plan should be based on the evaluation of results on a small scale. The plan should identify the important issues and the readiness gap elimination. The organization needs to assess the 5M readiness for IoT implementation evaluation. The implementation should focus on machine/setting accuracy, suitability for IoT, technology availability, developability, material availability, quality, thickness, the accuracy of measurements, standard operating procedure (S.O.P), display, tablet, mobile, etc. The implementation should also consider the graphical user interface (GUI) for data collection and analysis.

4.4.3. Evaluation & Improvement (Phase-3):

In this phase, the organization needs to evaluate the implementation results and identify areas for improvement. The organization needs to assess customer feedback, production performances, wastes, manual reports, damages, product rejection, profit and loss reports, quality, manpower feedback, etc. The evaluation should focus on plan vs. actual, MURI (Overburden), MURA (Unevenness), MUDA (Waste), and management support. The organization needs to assess potential failure, maintenance specialists, installation and connection, and sensors performance. The successful implementation confirms the new value/implementation."

"....The proposed framework can be used practically by organizations to assess their readiness for IoT, plan and implement an IoT system, evaluate the implementation results, and continuously improve the system. By leveraging this framework, organizations can leverage IoT technology to improve their operations, profitability, and customer satisfaction."

"Therefore, this study proposes a framework for IoT implementation that emphasizes proper management and communication of information throughout the material flow, which can help SMEs overcome challenges in implementing IoT and Industry 4.0 technologies. The implementation of proposed IoT framework eliminates the constraints of time and location, allowing for remote monitoring and control of the industry.

Therefore, it is suggested that SMEs should focus on incorporating more IoT technologies to fully benefit from its features and should consider proposed framework modification to make it compatible with Industry 4.0 and IoT technologies. "

  • In the performance section. The more details of the performance result should be analyzed and provided.

Response: Thank you for your comment. We made some changes to the paper to address this issue, which has been addressed in bullet point 2 (see Page 12 last Para. and Page 13 first half page).

We hope these changes satisfy your comment.

Reviewer 2 Report

Dear authors,

I think the paper under the title “Developing an IoT Framework for Industry 4.0 in Malaysian SMEs: An Analysis of Current Status, Practices, and Challenges” has the potential to be published in the journal Applied Science. However, certain changes should to be made:

1.      In the Introduction section it is necessary to explicitly state motivation for the research and research questions.

2.      Additionally, practical and methodological aims and contributions should be discussed.

3.      When it comes to the results of the study, statistically processed results and their interpretations are missing, this should be the main part of this study, without that part, the paper has no scientific significance.

4.      The discussion and conclusion should contain explicitly derived novelties and advantages, practical findings and implications, and finally research limitations.

The review of the literature is adequate, it contains relevant contemporary literature and the authors critical discussion. 

Author Response

Dear Reviewer 2.

We would like to thank you for your tme and effort to review our paper. The following are the response to your queries.

  •  In the Introduction section it is necessary to explicitly state motivation for the research and research questions.

Response: Thank you for your comment. This paragraph has been added in the Section.

"Industry 4.0 presents new technological capabilities by integrating emerging ICT technologies to optimize production performance. The adoption of Industry 4.0 technologies i.e., IoT is increasingly important and visible in manufacturing sectors. However, SMEs often lack the human, technical, and financial resources essential to implement Industry 4.0. ...."

  • Additionally, practical and methodological aims and contributions should be discussed.

Response: Thank you for your comment. These paragraphs have been added in the Discussion and Conclusion Sections.

"..The proposed framework can be used practically by organizations to assess their readiness for IoT, plan and implement an IoT system, evaluate the implementation results, and continuously improve the system. By leveraging this framework, organizations can leverage IoT technology to improve their operations, profitability, and customer satisfaction."

"Therefore, this study proposes a framework for IoT implementation that emphasizes proper management and communication of information throughout the material flow, which can help SMEs overcome challenges in implementing IoT and Industry 4.0 technologies. The implementation of proposed IoT framework eliminates the constraints of time and location, allowing for remote monitoring and control of the industry.

Therefore, it is suggested that SMEs should focus on incorporating more IoT technologies to fully benefit from its features and should consider proposed framework modification to make it compatible with Industry 4.0 and IoT technologies. "

  • When it comes to the results of the study, statistically processed results and their interpretations are missing, this should be the main part of this study, without that part, the paper has no scientific significance.

Response: Thank you for your comment. However, this study is qualitative. Qualitative methodology typically does not rely on statistical analysis for interpreting data. However, content analysis can provide data that can be analyzed using both qualitative (interpretive) and statistical approaches (Richard, 2013). But this research involves thematic analysis and statistical processes are not required for the data analysis.

We hope this satisfy your query.

  • The discussion and conclusion should contain explicitly derived novelties and advantages, practical findings and implications, and finally research limitations.

Response: Thank you for your comment. These paragraphs have been added in the Discussion and Conclusion Sections (see bullet point 2). Additional paragraph is as follows.

". This study framework has some implication as it can help Malaysian SMEs improve their manufacturing competitiveness and efficiency, overcome challenges in implementing IoT and Industry 4.0 technologies, and drive innovation and growth in the manufacturing sector. The theoretical implications of this research are that it can contribute to the existing literature on Industry 4.0 and IoT implementation. Furthermore, the proposed framework can help to establish new theoretical models and frameworks for future research on the implementation of Industry 4.0 technologies in the industrial/manufacturing sector. This study has some limitations as one limitation of the study is the small sample size comprised of five participants from three different SMEs, which may not be representative of the entire SMEs population in Malaysia. Additionally, the interviews were conducted online due to the COVID-19 pandemic, which may have affected the quality of the data collected compared to in-person interviews. Finally, the study's reliance on thematic analysis may have introduced researcher bias in the data analysis process. "

We hope these satisfy your queries.

Thank you.

Reviewer 3 Report

Accepted with following suggestions

1) This statement should be placed 1st

This qualitative study aimed to explore the current status, 14
practices, and challenges of Internet of Things (IoT) implementation and to develop an IoT framework for Industry 4.0 in Malaysia.

2) Introduction: Start is not good, authors put literature first. Introduce your research with need or urgency of research then present problem.

definations or literature should be place in next section

3) Literature is ok

4) Methods: In this section explain why you choose qualitative methods, define interviews its types and your choosen one ect

5) Discussion is ok

6) Conclusion: Add practical and theoratical implications

Author Response

Dear Reviewer 3,

Thank you for your time and effort to review our paper. The following are the response to your queries.

1) This statement should be placed 1st

This qualitative study aimed to explore the current status, 14
practices, and challenges of Internet of Things (IoT) implementation and to develop an IoT framework for Industry 4.0 in Malaysia.

Response: We have added this in the abstract as per suggestion.

2) Introduction: Start is not good, authors put literature first. Introduce your research with need or urgency of research then present problem. definations or literature should be place in next section

Response: We have made some changes as per suggestion. We have added this at the start of the Introduction to show the urgency of the work.

"Industry 4.0 presents new technological capabilities by integrating emerging ICT technologies to optimize production performance. The adoption of Industry 4.0 technologies i.e., IoT is increasingly important and visible in manufacturing sectors. However, SMEs often lack the human, technical, and financial resources essential to implement Industry 4.0. "

3) Literature is ok

Response: Thank you so much. We really appreciate it.

4) Methods: In this section explain why you choose qualitative methods, define interviews its types and your choosen one ect

Response: This has been corrected as suggested.

"It was carried out using a qualitative method through online interviews and a focus group. The qualitative method provides insights into the research objectives through primary data and helps to improve the ideas or hypotheses of the new framework's implementation [76]. Therefore, this study adopted the qualitative method. "

"According to O’Connor and Madge [78], online interviews offer greater convenience and cost-effectiveness compared to face-to-face interviews, with participants enjoying the flexibility to choose the time and location of their participation. "

5) Discussion is ok.

Response: Thank you so much. We really appreciate it.

6) Conclusion: Add practical and theoratical implications

Response: We have made some changes as per suggestion. We have added these paragraph in the Conclusions section. 

"Therefore, this study proposes a framework for IoT implementation that emphasizes proper management and communication of information throughout the material flow, which can help SMEs overcome challenges in implementing IoT and Industry 4.0 technologies. The implementation of proposed IoT framework eliminates the constraints of time and location, allowing for remote monitoring and control of the industry.

Therefore, it is suggested that SMEs should focus on incorporating more IoT technologies to fully benefit from its features and should consider proposed framework modification to make it compatible with Industry 4.0 and IoT technologies. "

"This study framework has some implication as it can help Malaysian SMEs improve their manufacturing competitiveness and efficiency, overcome challenges in implementing IoT and Industry 4.0 technologies, and drive innovation and growth in the manufacturing sector. The theoretical implications of this research are that it can contribute to the existing literature on Industry 4.0 and IoT implementation. Furthermore, the proposed framework can help to establish new theoretical models and frameworks for future research on the implementation of Industry 4.0 technologies in the industrial/manufacturing sector. This study has some limitations as one limitation of the study is the small sample size comprised of five participants from three different SMEs, which may not be representative of the entire SMEs population in Malaysia. Additionally, the interviews were conducted online due to the COVID-19 pandemic, which may have affected the quality of the data collected compared to in-person interviews. Finally, the study's reliance on thematic analysis may have introduced researcher bias in the data analysis process. "

We hope these satisfy your queries.

Round 2

Reviewer 2 Report

Paper has appropriate form for publication.